# Out of the Memory Barrier: A Highly Memory Efficient Training System for LLMs with Million-Token Contexts

**Wenhao Li**[*] **Daohai Yu**[*] **Gen Luo, Yuxin Zhang, Fei Chao & Rongrong Ji**[†]
Key Laboratory of Multimedia Trusted Perception and Efficient Computing
Ministry of Education of China, Xiamen University

**Yifan Wu**
Peking University

**Jiaxin Liu**
University of Illinois Urbana-Champaign

**Ziyang Gong**
Shanghai Jiao Tong University

**Zimu Liao**
Shanghai AI Laboratory

## Abstract

Training Large Language Models (LLMs) on long contexts is severely constrained by prohibitive GPU memory overhead, not training time. The primary culprits are the activations, whose memory footprints scale linearly with sequence length. We introduce OOMB, a highly memory-efficient training system that directly confronts this barrier. Our approach employs a chunk-recurrent training framework with on-the-fly activation recomputation, which maintains a constant activation memory footprint ($\mathcal{O}(1)$) and shifts the primary bottleneck to the growing KV cache. To manage the KV cache, OOMB integrates a suite of synergistic optimizations: a paged memory manager for both the KV cache and its gradients to eliminate fragmentation, asynchronous CPU offloading to hide data transfer latency, and page-level sparse attention to reduce both computational complexity and communication overhead. The synergy of these techniques yields exceptional efficiency. Our empirical results show that for every additional 10K tokens of context, the end-to-end training memory overhead increases by a mere 10MB for Qwen2.5-7B. This allows training Qwen2.5-7B with a 4M-token context on a single H200 GPU, a feat that would otherwise require a large cluster using context parallelism. This work represents a substantial advance in resource efficiency for long-context LLM training. The source code is available at https://github.com/wenhaoli-xmu/OOMB.

## 1 Introduction

Long-context modeling remains a formidable challenge in training Large Language Models (LLMs). The principal obstacle is not only training time but also the prohibitive GPU memory overhead associated with long sequences (Liu et al., 2025b; Wang et al., 2024). Research indicates that significant long-context capabilities can be achieved with a surprisingly small number of training steps (Peng et al., 2024; Ding et al., 2024), and that fine-tuning on as little as 5% of the original pretraining data may suffice for commercial-grade performance (Gao et al., 2025; Bai et al., 2024).

Despite this modest data requirement, the number of tokens processed in a single iteration grows dramatically. This causes GPU memory consumption from activations and the KV cache to scale linearly with context length, rapidly exhausting available resources. For instance, a model with a $4\times$ GQA (Ainslie et al., 2023) ratio requires 64GB for the KV cache alone when processing a 256K context. This demand overwhelms an A100 GPU before even accounting for other network activations, often making 32K tokens a practical limit for single-GPU training (Grattafiori et al., 2024; Yang et al.,

---

[*]Equal contribution
[†]Corresponding author

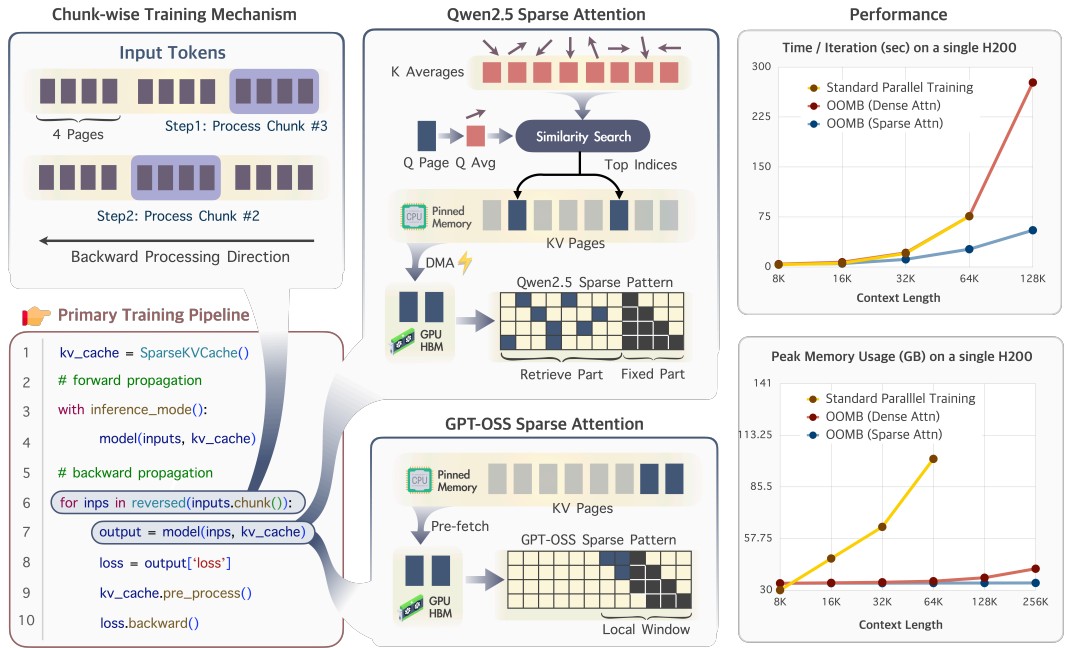

Figure 1: An overview of the OOMB training framework. *(Left)* OOMB processes sequences chunk-by-chunk with activation recomputation, maintaining a constant activation memory footprint ($\mathcal{O}(1)$) and shifting the bottleneck to the KV cache. *(Center)* The growing KV cache is managed through a combination of paged memory, asynchronous CPU offloading, and page-level sparse attention. *(Right)* Performance benchmarks show that the integrated system drastically reduces peak GPU memory, enabling efficient training on long contexts with minimal memory overhead.

2025). This immense memory pressure renders popular techniques like ZeRO3 (Rajbhandari et al., 2020) and tensor parallelism (Shoeybi et al., 2020) insufficient.

While various training-free methods for context extension exist (Jin et al., 2024; Han et al., 2024), their practical utility is limited. Prior work has established that common evaluation metrics like perplexity and targeted retrieval are unreliable indicators of true long-range reasoning (Gao et al., 2024; Bai et al., 2024). Achieving robust performance necessitates dedicated contiguous pretraining.

To this end, this paper directly confronts the central problem: how to train longer-context models with fewer resources without compromising performance. We introduce **O**ut **O**f the **M**emory **B**arrier (OOMB), a chunk-recurrent training framework that processes sequences in segments. During the forward pass, each chunk's activations are computed and then immediately discarded. For the backward pass, they are recomputed on-the-fly. This strategy maintains a constant activation memory footprint, regardless of the total sequence length. This approach, however, shifts the primary bottleneck to the KV cache, which must be retained throughout the training step and scales linearly with context length.

To overcome this challenge, OOMB integrates a system of co-designed optimizations for efficient KV cache management:

- **Paged KV Cache and Gradient Management.** We introduce a paged memory manager, inspired by (Kwon et al., 2023), for both the KV cache and its gradients. This design eliminates expensive memory operations and mitigates fragmentation when appending new key-value pairs.

- **Specialized Kernels.** We implement custom operators that execute all operations on the KV cache and its gradients directly, making them opaque to PyTorch's autograd system. This avoids storing the KV cache as an activation and allows for direct gradient accumulation.

- **Asynchronous KV Cache CPU Offloading.** We designed an asynchronous mechanism to preemptively offload the KV cache and its gradients, whose memory grows with context length, to CPU memory. The resulting data transfer latency is effectively masked by computation.

Table 1: Comparison of Long-Context Training Methods. OOMB achieves $\mathcal{O}(1)$ activation memory complexity, where $N$ denotes the context length. This key advantage allows OOMB, when combined with other techniques, to practically enable single-GPU training on contexts exceeding 4M tokens.

| Method | GPUs | Activation | Exact Attn | Max Tokens |
|---|---|---|---|---|
| SeCO (Li et al., 2025) | $1\times$H200 | $\mathcal{O}(1)$ | ✓ | 128K |
| LongLoRA (Chen et al., 2024) | | $\mathcal{O}(N)$ | ✗ | 128K |
| ZeRO3 Offload | $8\times$H200 | $\mathcal{O}(N)$ | ✓ | 128K |
| Ring Flash Attention | | $\mathcal{O}(N/8)$ | ✓ | 256K |
| Tensor Parallelism | | $\mathcal{O}(N/8)$ | ✓ | 256K |
| OOMB + Dense Attn | $1\times$H200 | $\mathcal{O}(1)$ | ✓ | 4M |
| OOMB + Sparse Attn | | $\mathcal{O}(1)$ | ✗ | 4M+ |

- **Page-level Sparse Attention.** Our paged memory architecture natively supports page-level sparse attention (Yuan et al., 2025; Lu et al., 2025). This reduces the computational complexity of attention and minimizes the communication overhead of our offloading mechanism.

The synergy of these techniques yields exceptional GPU memory efficiency. Our results show that for every additional 10K context tokens, the end-to-end training memory overhead increases by a mere 10MB for Qwen2.5-7B. This allows training Qwen2.5-7B with a 4-million-token context on a single H200 GPU. In contrast, context parallelism methods (Liu et al., 2024; Lin, 2025; Shoeybi et al., 2020) require large clusters for similar tasks (Xu et al., 2025), highlighting a substantial advance in resource efficiency.

## 2 RELATED WORK

**Efficient Long-Context Extension.** A core challenge in extending language models to long contexts is their failure to generalize to positional encodings not seen during pretraining (Han et al., 2024). While continued training can resolve this issue, the associated GPU memory costs are often prohibitive. This has motivated training-free context extension methods, which typically modify positional encodings to accommodate longer sequences (Chen et al., 2023; Han et al., 2024; Jin et al., 2024; An et al., 2024). However, these approaches introduce significant trade-offs. Strategies based on interpolation or extrapolation of positional encodings can degrade performance on short-text tasks by reducing the resolution of positional information (Gao et al., 2024). Others that reuse position indices achieve high scores on perplexity benchmarks but have been shown to fail on tasks requiring deep contextual understanding, indicating a superficial grasp of the extended context (Gao et al., 2024).

Given these limitations, state-of-the-art models still rely on fine-tuning to expand their context windows (Grattafiori et al., 2024; Yang et al., 2025; Liu et al., 2025a). While recent advances in parameter-efficient fine-tuning (PEFT) have successfully minimized the memory footprint of model weights and optimizer states (Zhao et al., 2025; Zhang et al., 2025; Zhong et al., 2025), they do not alleviate the activation memory bottleneck that scales linearly with sequence length. Consequently, the prevailing trend has shifted from algorithmic efficiency (Chen et al., 2024) toward overcoming memory barriers with massive computational resources (Liu et al., 2024; Li et al., 2024; Jacobs et al., 2024; Shoeybi et al., 2020). For instance, recent efforts have required a 256-GPU cluster to extend a model's context window to four million tokens (Xu et al., 2025).

**Serial vs. Parallel Training Paradigms.** Parallel training processes an entire sequence in a single pass, maximizing GPU utilization and scalability. However, its memory footprint scales linearly with sequence length, creating a significant bottleneck for long-context models. In contrast, serial training is highly memory-efficient, as it only activates a small part of the network at any given time, but at the cost of reduced parallelism.

For long-context fine-tuning, where dataset sizes are often moderate, memory consumption rather than throughput is the primary constraint (Peng et al., 2024; Gao et al., 2024). This observation motivates a hybrid approach that processes sequences in chunks: parallel within each chunk and serial between them. This strategy is standard in modern LLM inference engines (Kwon et al., 2023; Ye et al., 2025) where it incurs negligible latency. Its application to training, however, remains

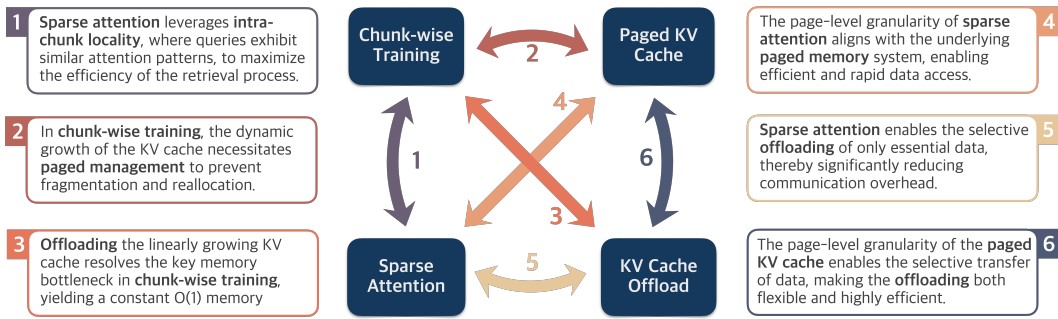

Figure 2: The synergistic architecture of OOMB's efficient KV cache management relies on four deeply interdependent core components: chunk-wise training, paged key-value cache, sparse attention, and key-value cache offload. Each optimization enables and enhances the others, collectively resolving the memory bottleneck of long-context training.

underdeveloped. An early exploration, SeCO (Li et al., 2025), introduced chunk-wise training but lacked critical optimizations at the operator and memory management levels. Consequently, it could only train a 16K context model on a single consumer-grade GPU. Under identical conditions, our fully-optimized system extends this capability by more than tenfold.

**Context Parallelism.** Context Parallelism (CP) and our proposed method both segment long sequences but diverge significantly in their scalability and overhead profiles. CP, inspired by Ring Attention (Liu et al., 2024) and implemented in frameworks like Megatron-LM (Shoeybi et al., 2020) and DeepSpeed Ulysses (Jacobs et al., 2024), distributes a single sequence across multiple GPUs. This design introduces substantial inter-GPU communication overhead that scales with the number of devices. In contrast, the overhead of our serial approach is limited to repeated KV cache accesses and the latency from CPU offloading. The trade-off is clear: CP excels with large GPU clusters and specialized high-bandwidth networks, whereas our method maintains a moderate and predictable overhead, eliminating the need for such extensive infrastructure.

## 3 PRELIMINARY: CHUNK-WISE TRAINING

The core strategy of OOMB is rooted in the principles of sequential processing, which have long been used to make training Recurrent Neural Networks (RNNs) memory efficient. We first revisit this foundation and then describe how it can be adapted to the Transformer architecture, setting the stage for the challenges that OOMB is designed to solve.

**Activation Recomputation in Sequential Models.** In a standard RNN, the forward pass iteratively computes an output $y_i$ and a new hidden state $m_i$ at each timestep $i$:

$$y_i, m_i, \mathbf{a_i} \leftarrow \mathcal{F}_{\text{RNN}}(x_i, m_{i-1}; \Theta) \tag{1}$$

where $\mathbf{a_i}$ represents the intermediate activations cached for the backward pass. For long sequences, storing all activations $\{\mathbf{a_1}, ..., \mathbf{a_T}\}$ creates a memory bottleneck that scales linearly with sequence length. Activation recomputation (Sohoni et al., 2022; Bencheikh et al., 2024) circumvents this by discarding $\mathbf{a_i}$ during the forward pass and regenerating it on-the-fly just before its use in the backward pass. This trades a modest amount of computation for a significant reduction in memory.

**Adapting Sequential Processing to Transformers.** While Transformers are inherently parallel, they can be trained using a similar chunk-wise, sequential paradigm. We partition a long input sequence into $S$ chunks, $\{X_1, X_2, ..., X_S\}$, and process them serially. The forward pass for chunk $i$ attends to the KV caches from all preceding chunks, which act analogously to an RNN's hidden state:

$$Y_i, M_i, \mathbf{A_i} = \mathcal{F}_{\text{Transformer}}(X_i, \{M_1, M_2, ..., M_{i-1}\}; \Theta) \tag{2}$$

Here, $M_i$ is the KV cache generated by chunk $i$, and $\mathbf{A_i}$ represents its intermediate activations. By applying the same activation recomputation strategy, we discard $\mathbf{A_i}$ after the forward step and

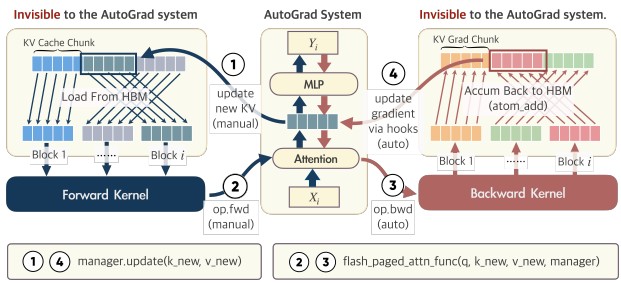
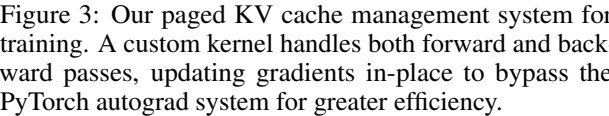
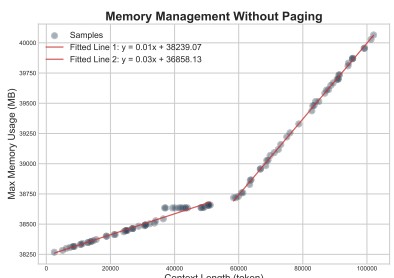

Figure 3: Our paged KV cache management system for training. A custom kernel handles both forward and backward passes, updating gradients in-place to bypass the PyTorch autograd system for greater efficiency.

Figure 4: A non-paged implementation can result in memory usage nearly $3\times$ higher than the theoretical requirement.

recompute it for the backward pass:

$$d\Theta, \{dM_1, ..., dM_{i-1}\} \leftarrow \text{backward}(dY_i, dM_i, \mathbf{A_i}) \tag{3}$$

This chunk-recurrent approach ensures that the activation memory footprint remains constant, determined only by the size of a single chunk. Consequently, this paradigm fundamentally shifts the memory bottleneck: the challenge is no longer the activations, but the management of the KV cache $\{M_1, ..., M_S\}$ and its gradients, which still grow linearly with the sequence length.

# 4 THE OOMB TRAINING SYSTEM

OOMB builds upon the chunk-wise training paradigm by integrating three synergistic components: paged KV cache management, sparse attention, and asynchronous KV cache offloading. As illustrated in Figure 2, these components are highly interdependent, and their combined effect is critical to the system's overall efficiency. This section provides a detailed description of each component.

## 4.1 PAGED MEMORY MANAGEMENT FOR KV CACHE AND GRADIENTS

Chunk-wise training requires the KV cache to grow incrementally, a process analogous to autoregressive decoding. Standard tensor concatenation for this purpose is inefficient, leading to frequent memory reallocations, copy operations, and severe memory fragmentation (Kwon et al., 2023). As shown in Figure 4, this inefficiency causes memory consumption to grow disproportionately with context length, limiting scalability.

To resolve this, OOMB incorporates a paged memory manager for both the KV cache and its gradients. Since existing paged attention solutions are designed for inference and lack backpropagation support, we developed a custom Triton kernel that handles both forward and backward passes (Figure 3).

Our implementation is designed to bypass the PyTorch `autograd` engine. This approach provides two key benefits. First, custom CUDA kernels accumulate gradients in-place using `atomic_add`, which eliminates intermediate buffers and reduces memory I/O. Second, decoupling the KV cache from the autograd graph prevents PyTorch from storing it as an activation. This design simplifies memory management and facilitates efficient offloading by maintaining a leaner computation graph.

## 4.2 SPARSE ATTENTION

The quadratic complexity of attention is a primary bottleneck in long-context training. OOMB addresses this by integrating page-level sparse attention, which reduces computational cost and is naturally supported by our paged KV cache architecture (Chen et al., 2024; Li et al., 2025). We implement two variants of sparse attention tailored to different model architectures.

**Approximating Dense Attention for Qwen2.5.** For models with dense attention like Qwen2.5 (Yang et al., 2025), we employ a Top-K page retrieval method to approximate the full attention pattern (Tang et al., 2024; Lu et al., 2025). The selected page indices from the forward pass are cached for reuse

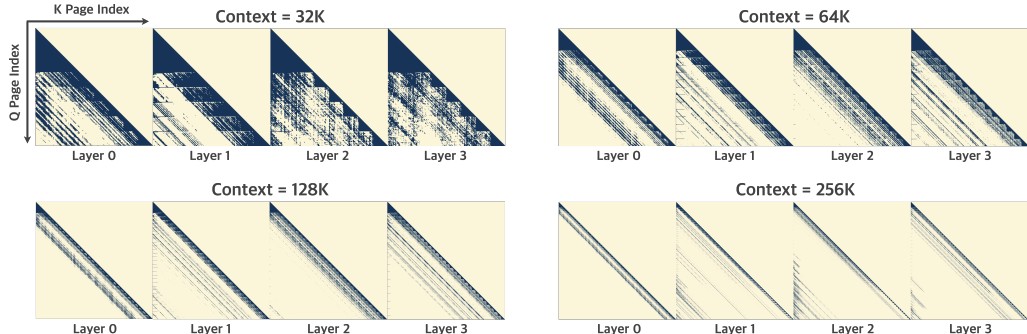

Figure 5: Visualization of the page-level sparse attention patterns for the Qwen2.5-7B model. Each subplot displays the retrieved key pages (x-axis) for each query page (y-axis) across the initial four layers at context lengths scaling from 32K to 256K.

during backpropagation. The retrieval process begins by computing a single representative vector $K_{\text{avg}} \in \mathbb{R}^{n \times D}$ for each of the $n$ key pages by averaging. Given queries $Q \in \mathbb{R}^{(m \times P) \times D}$ from the current chunk, where $m$ is the number of query pages in the chunk and $P$ is the page size (tokens per page), we compute similarity scores between each query token and all page-representative vectors. A voting mechanism then aggregates these scores for each of the $m$ query pages to identify the most relevant key pages:

$$\underset{\mathbb{R}}{\text{Score}}[i,j,k] \leftarrow Q[i,j]^\top K_{\text{avg}}[k], \quad \underset{m \times P \times n}{\text{Score}} \leftarrow \text{softmax}(\text{Score}), \quad \underset{m \times n}{\text{Score}} \leftarrow \text{sum}(\underset{m \times P \times n}{\text{Score}}). \quad (4)$$

This retrieval process, visualized in Figure 5, incurs negligible computational overhead with page sizes ($P$) of 128 on H200 GPUs.

**Native Sparse Attention for GPT-OSS.** For natively sparse models like GPT-OSS (Agarwal et al., 2025), which alternate between dense and local attention layers, our implementation is straightforward. The dense layers use the same Top-K retrieval method described above, while the local attention layers retrieve only the most recent KV pages.

## 4.3 ASYNCHRONOUS KV CACHE OFFLOADING

The KV cache is the only component in OOMB whose memory footprint scales with sequence length. To enable training on extremely long contexts, we introduce an asynchronous offloading mechanism that transfers the KV cache between GPU and CPU memory. The offloading strategy is adapted to the attention type to maximize the overlap between data transfer and computation.

**Offloading for Dense and Local Attention.** For these attention patterns, the required KV pages for a given layer are known in advance. We employ a pre-fetching strategy. During the forward pass for layer $i - 1$, the KV cache for layer $i$ is asynchronously fetched from the CPU. A symmetric process occurs during the backward pass. This approach effectively hides most of the data transfer latency, resulting in an end-to-end overhead lower than 5%.

**Offloading for Sparse Attention.** In retrieval-based sparse attention, the required KV pages are not known until the query vectors for the current layer have been computed. Consequently, during the forward pass, we initiate page retrieval and the corresponding asynchronous data transfer immediately after the query projection. The small volume of data required by sparse attention allows this transfer to be overlapped with the subsequent key and value projection computations. For the backward pass, the required page indices are already cached from the forward pass, allowing us to use the same pre-fetching strategy as in dense attention.

Both offloading schemes, illustrated in Figure 6, are implemented using pinned CPU memory and dedicated CUDA streams to ensure true asynchronicity. Data transfers are handled by the GPU's Direct Memory Access (DMA) engine to minimize latency.

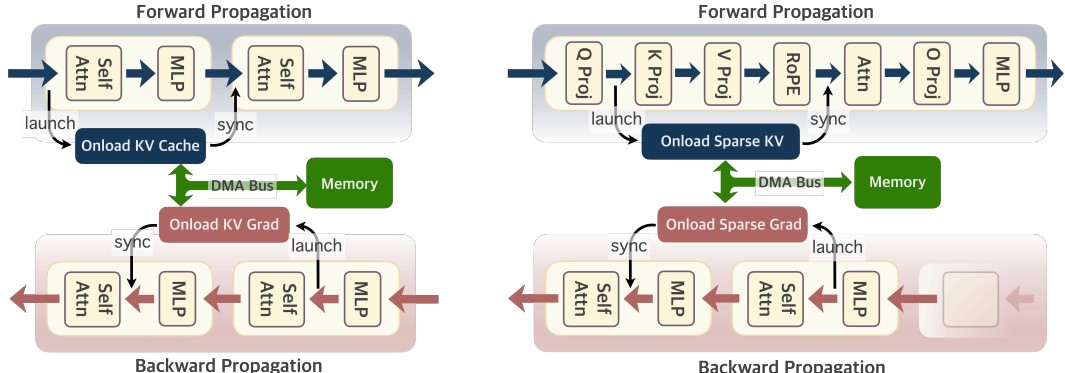

Figure 6: Asynchronous KV cache offloading in OOMB. *(Left)* For dense and local attention, the KV cache for the next layer is pre-fetched during computation of the current layer. *(Right)* For sparse attention, the forward pass overlaps data transfer with key-value projections, while the backward pass reverts to the pre-fetching strategy.

## 5 EXPERIMENTS

### 5.1 EXPERIMENTAL SETUP

Unless otherwise specified, all experiments adhere to the following configuration.

**Model and Dataset.** We use the Qwen2.5-7B model (Yang et al., 2025), a powerful foundation model, for all efficiency and scalability benchmarks. The training data is sourced from the arXiv dataset (Soboleva et al., 2023). To construct sequences of the required lengths for our long-context experiments, we concatenate samples from the dataset.

**Baseline Methods.** For efficiency evaluation, we compare our method against two baselines: (1) a standard parallel training setup using FlashAttention (Dao, 2024) and layer-wise gradient checkpointing; and (2) Ring Flash Attention (RFA) (Lin, 2025), a state-of-the-art implementation of Ring Attention (Liu et al., 2024).

Additional details are provided in Appendix B.

### 5.2 MEMORY, TIME, AND SCALABILITY ANALYSIS

We evaluate OOMB's efficiency against two paradigms: standard parallel training and context parallelism.

**Comparison with Parallel Training.** Table 2 presents a comprehensive efficiency comparison between OOMB and a parallel training baseline using FlashAttention. Our analysis highlights several key findings:

- At a 32K context length, OOMB with dense attention already outperforms the baseline, and this performance advantage widens as the context length increases.

- The asynchronous offloading mechanism imposes negligible latency overhead on dense attention, demonstrating its high efficiency.

- Sparse attention provides substantial acceleration on longer sequences, yielding speedups of approximately $3.7\times$ at 64K and up to $13.5\times$ at 256K.

- For sparse attention with offloading, a larger chunk size mitigates latency; a 12K chunk size, for example, adds only a 12% overhead.

- Our offloading system dramatically improves memory scalability for both dense and sparse attention, keeping peak memory usage nearly constant irrespective of context length.

- Even without CPU offloading, OOMB reduces the memory growth rate by more than fivefold compared to the parallel training baseline.

Table 2: Performance and Ablation of OOMB for Qwen2.5-7B on an H200 GPU. Per-iteration latency and peak memory for various OOMB configurations (ablating chunk size, token budget, gradient checkpointing, and CPU offload) and a FlashAttention (Dao, 2024) baseline. Red intensity indicates: (1) for latency, the slowdown vs. the Ckpt-only counterpart; (2) for memory, the growth rate as context length increases. Single-GPU offload results are reported under full 8-GPU system load.

| Method | Budget | Ckpt. | Offload | 8K | 16K | 32K | 64K | 128K | 256K | 8K | 16K | 32K | 64K | 128K | 256K |
|---|---|---|---|---|---|---|---|---|---|---|---|---|---|---|---|
| | | Configuration | | Per-iteration Latency (s) | | | | | | Peak Memory Usage (MB) | | | | | |
| Baseline | - | ✓ | - | 2.4 | 5.7 | 20.4 | 76.1 | OOM | OOM | 30298 | 47025 | 64985 | 1002546 | OOM | OOM |
| ▼ *Chunk Size = 4K* | | | | | | | | | | | | | | | |
| OOMB + Dense Attn | - | ✓ | ✗ | 2.6 | 6.0 | 19.3 | 68.9 | 260.6 | 1015.9 | 34310 | 36326 | 39804 | 44390 | 55143 | 76650 |
| | | ✗ | ✓ | 2.4 | 6.0 | 18.6 | 63.4 | 233.9 | 899.6 | 57369 | 57529 | 57743 | 58175 | 59065 | 60788 |
| | | ✓ | ✓ | 3.0 | 6.7 | 20.7 | 70.7 | 260.4 | 985.8 | 33686 | 33848 | 34076 | 34504 | 35404 | 37129 |
| OOMB + Sparse Attn | 4K+512 | ✓ | ✗ | 2.7 | 4.4 | 8.8 | 18.0 | 36.6 | 75.7 | 34317 | 36335 | 39804 | 44409 | 55177 | 76719 |
| | | ✗ | ✓ | 3.0 | 4.5 | 9.5 | 19.6 | 40.0 | 83.3 | 57348 | 57372 | 57377 | 57380 | 57410 | 57448 |
| | | ✓ | ✓ | 2.8 | 5.2 | 10.8 | 22.6 | 46.5 | 96.9 | 33669 | 33694 | 33999 | 33712 | 33732 | 33769 |
| | 4K+2048 | ✓ | ✗ | 2.7 | 4.9 | 10.6 | 21.9 | 44.9 | 93.2 | 34317 | 36335 | 39827 | 44411 | 55178 | 76723 |
| | | ✗ | ✓ | 2.9 | 5.5 | 11.5 | 24.5 | 51.4 | 107.2 | 57348 | 57401 | 57413 | 57456 | 57479 | 57515 |
| | | ✓ | ✓ | 3.1 | 6.2 | 13.4 | 28.6 | 60.9 | 127.5 | 33669 | 33722 | 33735 | 33778 | 33800 | 33837 |
| | 4K+8192 | ✓ | ✗ | 2.6 | 5.9 | 14.3 | 31.5 | 66.2 | 138.1 | 34317 | 36335 | 39826 | 44413 | 55183 | 76733 |
| | | ✗ | ✓ | 2.4 | 6.7 | 16.2 | 37.0 | 83.4 | 178.6 | 57348 | 57421 | 57487 | 57614 | 57647 | 57752 |
| | | ✓ | ✓ | 2.9 | 7.4 | 18.2 | 42.8 | 94.0 | 201.4 | 33669 | 33743 | 33809 | 33857 | 34083 | 34083 |
| | 4K+32768 | ✓ | ✗ | 2.5 | 6.0 | 19.3 | 61.1 | 148.1 | 324.9 | 34317 | 36335 | 39826 | 44416 | 55196 | 76768 |
| | | ✗ | ✓ | 2.4 | 6.4 | 20.6 | 67.1 | 154.7 | 392.1 | 57348 | 57421 | 57520 | 57716 | 57716 | 58238 |
| | | ✓ | ✓ | 3.0 | 7.2 | 22.7 | 79.8 | 187.7 | 427.4 | 33669 | 33743 | 33842 | 34036 | 34426 | 34503 |
| ▼ *Chunk Size = 8K* | | | | | | | | | | | | | | | |
| OOMB + Dense Attn | - | ✓ | ✗ | 2.8 | 5.7 | 19.0 | 68.4 | 259.0 | 1014.5 | 31628 | 40795 | 43483 | 48860 | 59613 | 81119 |
| | | ✗ | ✓ | 2.3 | 5.5 | 17.6 | 64.3 | 237.4 | 922.2 | 71025 | 85705 | 85898 | 86282 | 87051 | 86589 |
| | | ✓ | ✓ | 3.0 | 6.1 | 19.7 | 70.6 | 264.3 | 1021.4 | 30374 | 38293 | 38485 | 38870 | 39639 | 41177 |
| OOMB + Sparse Attn | 8K+512 | ✓ | ✗ | 2.5 | 5.1 | 11.4 | 24.1 | 49.6 | 101.9 | 31826 | 40796 | 43488 | 48872 | 59639 | 81181 |
| | | ✗ | ✓ | 2.2 | 5.5 | 12.1 | 25.3 | 52.2 | 107.8 | 70979 | 85602 | 85624 | 85660 | 85694 | 85726 |
| | | ✓ | ✓ | 2.5 | 6.3 | 13.4 | 28.7 | 59.9 | 123.7 | 30380 | 38193 | 38215 | 38253 | 38284 | 38270 |
| | 8K+2048 | ✓ | ✗ | 2.5 | 5.4 | 12.4 | 26.3 | 54.4 | 111.6 | 31826 | 40797 | 43489 | 48873 | 59640 | 81183 |
| | | ✗ | ✓ | 2.4 | 5.4 | 12.6 | 27.9 | 58.2 | 120.7 | 70979 | 85610 | 85635 | 85716 | 85732 | 85757 |
| | | ✓ | ✓ | 2.6 | 6.6 | 14.7 | 31.9 | 67.3 | 142.1 | 30380 | 38201 | 38226 | 38262 | 38317 | 38364 |
| | 8K+8192 | ✓ | ✗ | 2.7 | 5.8 | 16.0 | 36.6 | 78.1 | 162.6 | 31826 | 40797 | 43489 | 48874 | 59646 | 81192 |
| | | ✗ | ✓ | 2.4 | 5.7 | 16.3 | 38.7 | 85.2 | 183.4 | 70979 | 85613 | 85695 | 85813 | 85952 | 85990 |
| | | ✓ | ✓ | 3.0 | 6.4 | 17.9 | 41.7 | 94.0 | 204.7 | 30380 | 38204 | 38286 | 38319 | 38427 | 38608 |
| | 8K+32768 | ✓ | ✗ | 2.8 | 5.7 | 18.9 | 63.1 | 159.8 | 345.6 | 31826 | 40797 | 43488 | 48877 | 59658 | 81227 |
| | | ✗ | ✓ | 2.4 | 5.7 | 18.9 | 64.8 | 165.0 | 355.2 | 70979 | 85613 | 85713 | 85917 | 86122 | 86122 |
| | | ✓ | ✓ | 3.0 | 6.4 | 20.8 | 70.8 | 177.7 | 408.5 | 30380 | 38204 | 38304 | 38508 | 38731 | 38856 |
| ▼ *Chunk Size = 12K* | | | | | | | | | | | | | | | |
| OOMB + Dense Attn | - | ✓ | ✗ | 2.8 | 6.1 | 19.6 | 70.9 | 262.5 | 1024.0 | 31627 | 44589 | 46605 | 52654 | 62735 | 84913 |
| | | ✗ | ✓ | 2.3 | 5.7 | 18.9 | 68.2 | 258.7 | 1017.2 | 71306 | 115169 | 114398 | 115190 | 117984 | 117984 |
| | | ✓ | ✓ | 2.8 | 6.1 | 19.6 | 70.9 | 262.5 | 1024.0 | 31575 | 42705 | 42879 | 44081 | 45858 | 45858 |
| OOMB + Sparse Attn | 12K+512 | ✓ | ✗ | 2.7 | 4.7 | 12.5 | 28.5 | 60.5 | 125.3 | 31626 | 44589 | 46609 | 52064 | 62760 | 84973 |
| | | ✗ | ✓ | 2.4 | 4.7 | 12.3 | 28.6 | 60.3 | 126.3 | 70979 | 113753 | 113765 | 113810 | 113881 | 113890 |
| | | ✓ | ✓ | 2.9 | 5.4 | 14.0 | 32.8 | 72.8 | 146.4 | 30379 | 42639 | 42681 | 42698 | 42737 | 42774 |
| | 12K+2048 | ✓ | ✗ | 2.8 | 4.9 | 13.5 | 31.3 | 66.7 | 138.1 | 31626 | 44589 | 46609 | 52065 | 62761 | 84975 |
| | | ✗ | ✓ | 2.4 | 5.1 | 13.9 | 32.8 | 69.8 | 146.7 | 70979 | 113753 | 113817 | 113880 | 113923 | 113963 |
| | | ✓ | ✓ | 3.0 | 5.7 | 15.2 | 36.4 | 78.9 | 163.5 | 30379 | 42609 | 42703 | 42766 | 42809 | 42845 |
| | 12K+8192 | ✓ | ✗ | 2.8 | 5.7 | 17.6 | 43.4 | 90.6 | 188.7 | 31626 | 44593 | 46609 | 52067 | 62765 | 84985 |
| | | ✗ | ✓ | 2.4 | 5.8 | 18.4 | 43.6 | 90.5 | 197.9 | 70979 | 113753 | 113830 | 113997 | 114017 | 114138 |
| | | ✓ | ✓ | 3.0 | 6.6 | 21.5 | 48.2 | 105.1 | 224.1 | 30379 | 42639 | 42716 | 42860 | 42968 | 43018 |
| | 12K+32768 | ✓ | ✗ | 2.8 | 5.7 | 18.9 | 66.4 | 170.0 | 373.9 | 31626 | 44589 | 46609 | 52070 | 62778 | 85019 |
| | | ✗ | ✓ | 2.4 | 5.7 | 18.5 | 67.2 | 169.0 | 385.3 | 70979 | 113753 | 113809 | 114028 | 114284 | 114558 |
| | | ✓ | ✓ | 3.1 | 6.5 | 20.6 | 74.0 | 185.6 | 421.2 | 30379 | 42639 | 42715 | 42944 | 43204 | 43443 |

Table 3: Per-device training throughput (tokens/sec) comparison against context-parallel methods. All OOMB configurations use a 4K chunk size with checkpointing and CPU offload; the sparse attention version adds an 8192 retrieval budget. TP: Tensor Parallel.

| Context | Ring Attn (report) | | RFA | OOMB (Dense Attn) | | OOMB (Sparse Attn) | |
|---|---|---|---|---|---|---|---|
| | 8×A100 | 16×TPUv4 | 4×H200 | 1×H200 | 4×H200 (TP) | 1×H200 | 4×H200 (TP) |
| 64K | - | - | 872.51 | 936.22 | 933.84 | 1560.38 | 1542.85 |
| 128K | - | - | 463.37 | 504.12 | 500.26 | 1394.16 | 1371.41 |
| 256K | 50 | 49 | 218.41 | 266.13 | 261.47 | 1301.60 | 1265.59 |

**Comparison with Context Parallelism Methods.** We also benchmarked OOMB against Ring Flash Attention (RFA) (Lin, 2025), a state-of-the-art implementation of Context Parallelism. This paradigm manages memory by splitting sequences across GPUs but introduces significant inter-device communication overhead. As shown in Table 3, OOMB achieves higher per-device training throughput, outperforming the heavily optimized RFA.

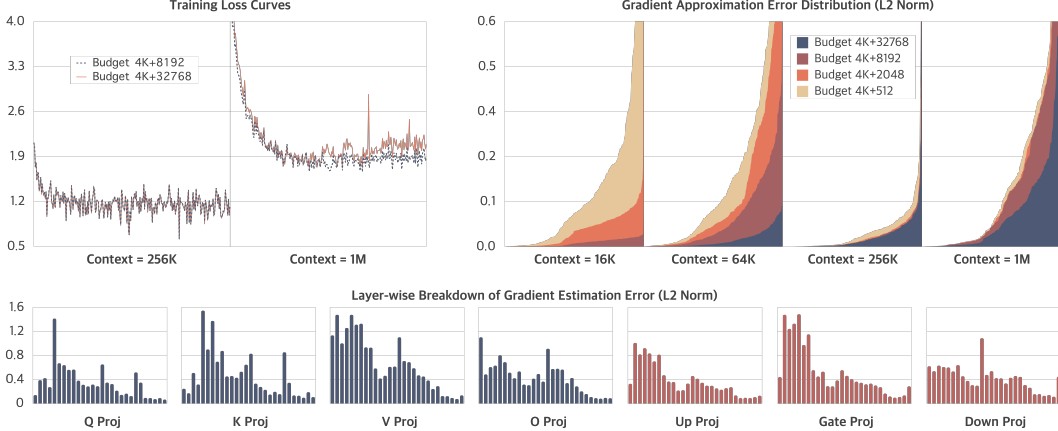

Figure 7: *(Top-left)* Training loss curves for Qwen2.5-7B (Yang et al., 2025) with 256K and 1M context lengths, employing sparse attention with retrieval budgets of 4K+8192 and 4K+32768. *(Top-right)* Distribution of the L2 norm of gradient approximation error for sparse attention across various context lengths (16K to 1M) and retrieval budgets. *(Bottom)* Layer-wise L2 norm of the gradient estimation error for different projection matrices, analyzed with a 64K context length and a 4K+8192 retrieval budget. Our work offers a preliminary validation of this sparse training; for more optimal sparse attention strategies, we refer readers to the established literature.

## 5.3 ACCURACY VALIDATION OF SPARSE ATTENTION

We empirically validated the accuracy of our sparse attention mechanism and its impact on model training. First, we analyzed the gradient approximation error for Qwen2.5-7B (Yang et al., 2025) across 16 configurations, composed of 4 context lengths and 4 retrieval budgets. As shown in Figure 7 *(Top-right)*, within the model's 128K native context limit, a larger retrieval budget results in a smaller approximation error. Beyond this limit, the error differences between budgets become less significant. The loss values for these settings, presented in Table 4a, further indicate that sparse attention does not substantially degrade model performance.

Next, we performed training runs at 256K and 1M context lengths using 8K and 32K retrieval budgets. The results in Figure 7 *(Top-left)* show that at 256K, both budgets yield rapid convergence to a loss below 1.2 with highly similar loss curves. At a 1M context, the 32K budget exhibited minor instability late in training, an effect we attribute to the learning rate and limitations in positional encoding extrapolation. Nevertheless, both configurations converged successfully to low loss values. As detailed in Table 4b, this approach is practical, allowing a 1M-context model to be trained for 256 steps in approximately 10 hours on a 4×H200 system with tensor parallelism.

Collectively, these findings confirm that sparse attention is a viable and effective method for long-context training, as it preserves model performance while achieving a constant $\mathcal{O}(1)$ memory footprint.

Table 4: (a) Training loss for the 16 configurations (4 context lengths × 4 sparse budgets) in the top-right subplot of Figure 7. (b) Resource usage during training for the 4 configurations (2 context lengths × 2 sparse budgets) in the top-left subplot of Figure 7. These experiments were conducted on four H200 GPUs using tensor parallelism, without gradient checkpointing or CPU offload.

(a) Loss values under different settings

| Context | Dense Attention | Sparse Attention | | | |
|---|---|---|---|---|---|
| | | 4K+512 | 4K+2048 | 4K+8192 | 4K+32768 |
| 16K | 1.32031 | 1.50000 | 1.36718 | 1.32031 | 1.32031 |
| 64K | 1.22387 | 1.47656 | 1.32812 | 1.21875 | 1.21875 |
| 256K | 2.21875 | 2.34375 | 2.28125 | 2.21875 | 2.21875 |
| 1M | 4.31250 | 4.31250 | 4.50000 | 4.71875 | 4.78125 |

(b) Resource usage (4×H200 tensor parallelism)

| Metric | 4K+8192 | | 4K+32768 | |
|---|---|---|---|---|
| | 256K | 1M | 256K | 1M |
| Latency/Iter (s) | 43.454 | 176.305 | 87.340 | 422.383 |
| Peak Memory (MB) | 35637 | 67944 | 35671 | 68145 |

## 6    CONCLUSION AND LIMITATIONS

We introduced OOMB, a highly memory-efficient training system that overcomes the primary memory barrier in long-context LLM training. By integrating a chunk-recurrent framework with activation recomputation, a paged KV cache manager, asynchronous CPU offloading, and page-level sparse attention, OOMB maintains a nearly constant memory footprint regardless of sequence length. Our results demonstrate that OOMB enables training a Qwen2.5-7B model with a 4-million-token context on a single H200 GPU, a task that would otherwise require a large-scale cluster. This work represents a significant advance in making long-context model training more accessible and resource-efficient.

Despite its efficiency, OOMB has several limitations. The chunk-wise serial processing introduces a latency overhead compared to fully parallel training paradigms, which may be a consideration for throughput-sensitive training scenarios. Additionally, our use of sparse attention is an approximation of the full attention mechanism, and its impact on model performance for tasks requiring dense global context requires further investigation. The effectiveness of the asynchronous offloading is also dependent on the available CPU-GPU interconnect bandwidth.

## ETHICS STATEMENT

The primary contribution of this work is a system designed to improve the computational efficiency of training large language models on long contexts. By significantly reducing the hardware requirements for such tasks, our research aims to democratize access to long-context model development for researchers and institutions with limited computational resources. A direct positive implication of this efficiency is the potential reduction in the overall energy consumption and carbon footprint associated with training these models, promoting more sustainable AI research practices.

We acknowledge the potential for dual use, as is common with foundational AI research. The tools we develop could be applied to train models for malicious purposes, such as generating misinformation. While this work focuses on the system's technical aspects, we advocate for the responsible development and deployment of models trained using our framework. Our experiments were conducted on the publicly available arXiv dataset, which consists of scientific literature and does not contain personally identifiable or sensitive information.

## REPRODUCIBILITY STATEMENT

To facilitate the reproducibility of our findings, we have made the complete source code for the OOMB training system available in the anonymous repository cited in the abstract. A detailed description of our experimental setup, including model specifications, datasets, hyperparameters, and evaluation protocols, is provided in Section 5.1 and Appendix B. The core methodology and its components are thoroughly explained in Section 4. We believe these resources offer a sufficient basis for researchers to verify our results and extend our work.

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

## A  USE OF LARGE LANGUAGE MODELS (LLMs)

During the writing of this paper, we utilized LLM solely for language editing to improve clarity and readability. We critically reviewed and revised all AI-generated suggestions to ensure the final text accurately reflects our original intent. All intellectual contributions, including the research design, methodology, analysis, and conclusions, are our exclusive work, and we take full responsibility for the academic integrity of this publication.

## B  EXPERIMENTAL DETAILS

All experiments were conducted using full-parameter fine-tuning in `bfloat16` precision. We employed the Adam optimizer with a learning rate of $5 \times 10^{-5}$ and betas of $(0.9, 0.98)$. To isolate the impact of sequence length on system performance, the per-GPU batch size was set to one, and all other model and training hyperparameters were maintained at their default settings. For OOMB-specific configurations, the KV cache page size was set to 128 for H200 GPUs.

System performance was evaluated using two primary metrics: peak GPU memory and per-iteration latency. Peak memory per device was recorded using PyTorch's `max_memory_allocated()` function, while latency was measured precisely with CUDA events. To ensure reliability and mitigate measurement noise, all reported results represent the minimum value from three independent runs.

## C  ADDITIONAL EXPERIMENTS AND ABLATION STUDIES

This section provides supplementary experiments that further validate the performance, scalability, and correctness of the OOMB framework.

### PERFORMANCE AND ABLATION STUDIES

**Scalability to 4M+ Tokens.** A key objective of OOMB is to enable training on sequence lengths that are infeasible with conventional methods on limited hardware. We benchmarked the per-iteration training time and memory usage of OOMB on contexts up to 8 million tokens. Figure 8 shows that OOMB maintains exceptional memory efficiency even at this scale. When using sparse attention, the training time scales in a near-linear fashion, making ultra-long context training practically achievable.

**Ablation on Chunk Size.** The chunk size is a key hyperparameter in OOMB that balances computational parallelism and activation memory. We evaluated the effect of varying the chunk size from 512 to 4096. As shown in Figure 9, larger chunk sizes improve computational efficiency by enabling better hardware utilization. However, these performance gains diminish beyond a size of

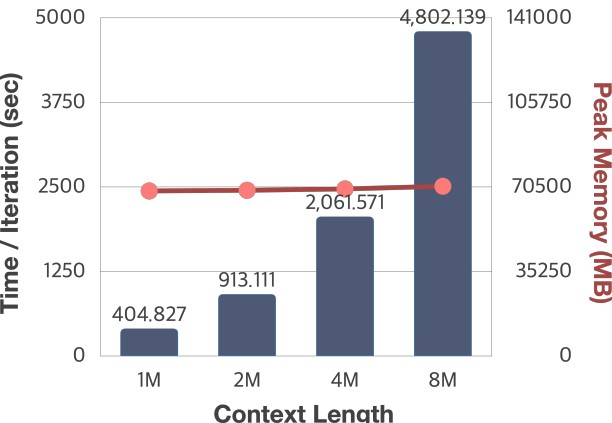

Figure 8: **Scalability to 4M+ Tokens.** OOMB demonstrates perfect memory efficiency up to a 8-million-token context length. With sparse attention, training time scales more linearly compared to dense attention. This experiment was conducted on 4×H200 GPUs using tensor parallelism.

4096. Memory consumption remains stable across all tested sizes. This analysis supports 4096 as a well-balanced default value, as it provides strong computational performance without excessive memory usage for activations.

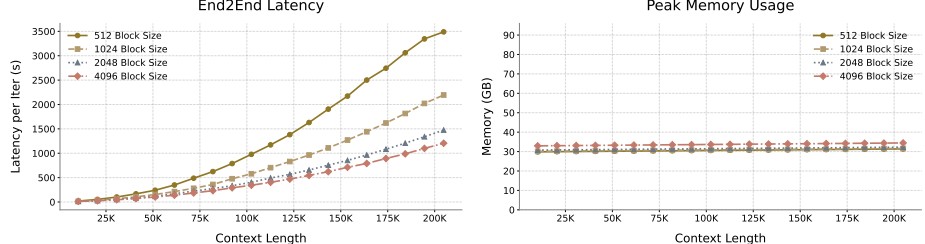

Figure 9: **Ablation on Chunk Size.** Increasing the chunk size improves computational throughput with diminishing returns, while peak memory usage remains stable. A chunk size of 4096 offers a strong balance between efficiency and memory. This experiment was conducted on a single A100 80G GPU.

