# OpenReview forum: "Out of the Memory Barrier: A Highly Memory-Efficient Training System for LLMs with Million-Token Contexts"
_ICLR.cc/2026/Conference — ICLR 2026 Poster_

### Official Review · Reviewer_qtG7 · 2025-10-19

**Soundness:** 2
**Presentation:** 3
**Contribution:** 1
**Rating:** 4
**Confidence:** 3

**Summary:**

The paper presents OOMB, a system for memory-efficient training of LLMs on long contexts. The approach combines chunk-recurrent training with activation recomputation, paged KV cache management, asynchronous CPU offloading, and sparse attention. The authors claim to enable training of Qwen2.5-7B with 4M-token contexts on a single H200 GPU.

**Strengths:**

1. The paper addresses an Important Problem: GPU memory is indeed a critical bottleneck for long-context LLM training, making this a relevant research direction.

2. The integration of multiple techniques (paging, offloading, sparse attention, custom kernels) shows good systems engineering.

3. Strong Memory Efficiency: The claimed 10MB per 10K tokens overhead is impressive if validated.

 4. Clear Presentation: The paper is generally well-written with good use of figures to illustrate the approach.

**Weaknesses:**

The paper demonstrates memory efficiency but never validates that the trained models actually work. In particular, the paper did not provide evaluations on long-context benchmarks (LongBench, RULER, BABILong, InfiniteBench), comparisons of final model quality between dense and sparse attention. Meanwhile the training curves stop at ~250 steps with loss ~1.2-4.7, and no evidence of convergence, and only used synthetic data. It would be great if the authors can provide complete training runs with evaluation on established benchmarks, and compare dense vs. sparse attention on major tasks.

The paper claimed that it can "training Qwen2.5-7B with a 4M-token context on a single H200 GPU". However, figure 8 (4M experiment) uses 4×H200 with tensor parallelism, not single GPU, 4M requires sparse attention (Table 2 shows dense attention OOMs at 256K on single GPU), and table 1 lists "OOMB + Dense Attn" at 4M—contradicts the data. Can the author clarify w.r.t this matter?

The paper has limited technical novelty, as each component exists: chunk-wise training: Standard for RNNs, used by SeCO for transformers; paged KV cache: directly from vLLM; CPU offloading: common practice; Sparse attention: well-established.

**Questions:**

Table 3 shows OOMB is only marginally faster than Ring Flash Attention per-device. When is OOMB preferable vs. context parallelism on a cluster?

---

> ### Author Response · Authors · 2025-11-18
>
> **[Q] Validates that the trained models actually work**
>
> [A] We sincerely appreciate the reviewer's constructive suggestion. To validate the effectiveness of our approach, we conducted additional evaluations on the RULER benchmark. As shown in the table below, the model trained via our OOMB strategy successfully handles the 256K context window. It achieves performance comparable to standard parallel training and significantly outperforms the original 128K baseline (which fails at 256K). This confirms that our method ensures model convergence and validity.
>
> | niah_multikey_1 | 128K | 256K |
> | --- | --- | --- |
> | Qwen2.5-7B-128K | 26.2 | 0.0 (Nulls 500/500) |
> | Qwen2.5-7B-256K (OOMB Trained) | 24.4 | 21.6 |
> | Qwen2.5-7B-256K (Parallel Trained) | 24.2 | 21.6 |

---

> ### Author Response · Authors · 2025-11-18
>
> **[Q] Comparisons of final model quality between dense and sparse attention**
>
> [A] Thank you for this insightful question. We would like to clarify that our work focuses on the system architecture, and our core contributions are orthogonal to the specific choice of sparse attention algorithm or position extrapolation methods.
>
> Consequently, we employed a naive sparse attention mechanism (naive top-k) solely for our efficiency tests. This approach allows us to simplify non-essential components and focus on the architectural validation.
>
> Regarding the model quality comparison between dense and sparse attention, we have gathered several relevant studies for your reference:
>
> * [1] Figure 4 and Table 1
> * [2] Figure 3, which provides scaling laws comparing dense and sparse attention.
> * [3] Table 2 and Figure 3, which demonstrate the comparison against dense attention in both text and video domains.
> * [4] Table 2, which presents a method highly similar to ours, differing only by the addition of a trainable projector.
>
> Notably, all the methods above are page-level, making them readily integrable with our framework (as our retrieval strategy is implemented in PyTorch). In fact, our repository already includes an implementation of NSA in addition to the naive top-k baseline.
>
> *Reference*
>
> [1] Jingyang Yuan, et al. Native Sparse Attention: Hardware-Aligned and Natively Trainable Sparse Attention. In ACL, 2025.
>
> [2] Enzhe Lu, et al. MoBA: Mixture of Block Attention for Long-Context LLMs. arXiv preprint arXiv:2502.13189, 2025.
>
> [3] Ruyi Xu, et al. XAttention: Block Sparse Attention with Anti-diagonal Scoring. In ICML, 2025.
>
> [4] Yizhao Gao, et al. SeerAttention: Learning Intrinsic Sparse Attention in Your LLMs. arXiv preprint arXiv:2410.13276, 2025.

---

> ### Author Response · Authors · 2025-11-18
>
> **[Q] Meanwhile the training curves stop at ~250 steps with loss ~1.2-4.7, and no evidence of convergence**
>
> [A] Thank you for your very careful observation. We appreciate the opportunity to clarify our results.
>
> We have respectfully double checked Figure 7 but seem to have difficulty locating the 1.2-4.7 loss range you mentioned. For instance, the y-axis in this figure has an upper bound of 4.0. We are concerned we may have misunderstood which part of the figure this refers to.
>
> From our perspective, the curves for 256K and 1M training show the loss dropping rapidly and consistently remaining below 2.5, which we interpret as evidence of convergence.
>
> To ensure we address your concern accurately, we are wondering if you might be referring to the late-stage instability observed in the '4K+32768' combination during the 1M context training? We would be very grateful for any clarification on this point, as it would help us improve the paper.

---

> ### Author Response · Authors · 2025-11-18
>
> **[Q] The paper claimed that it can "training Qwen2.5-7B with a 4M-token context on a single H200 GPU". However, figure 8 (4M experiment) uses 4×H200 with tensor parallelism, not single GPU, 4M requires sparse attention (Table 2 shows dense attention OOMs at 256K on single GPU), and table 1 lists "OOMB + Dense Attn" at 4M—contradicts the data.**
>
> [A] Thank you again for your meticulous review. We appreciate the opportunity to clarify the data regarding Table 2.
>
> Concerning the assertion that ”Table 2 shows dense attention OOMs at 256K on single GPU,” we have respectfully re-examined our results but seem to have difficulty locating this specific finding.
>
> Our data in Table 2 indicates that dense attention at a 256K context requires only 37,129MB. This is well within the 141GB memory capacity of a single H200 GPU. To further illustrate this, the data below shows the peak memory for *dense attention* remains well within the H200's limit, even as the context scales significantly larger:
>
>
> | 512K | 1M | 2M | 4M |
> | --- | --- | --- | --- |
> | 40579MB | 47437MB | 61225MB | (Experiment in Progress) |
>
> The 4M context experiment is currently running, and we will be sure to provide this result as soon as it is available. We hope this data helps resolve the concern, and we are very happy to provide any further clarification you may need.

---

> ### Author Response · Authors · 2025-11-18
>
> **[Q] When is OOMB preferable vs. context parallelism on a cluster?**
>
> [A] That is an insightful question. The choice between OOMB and CP is indeed scenario-dependent, as each has distinct
>
> First, the performance of CP is highly contingent on communication bandwidth. For extremely long context lengths, inter-node communication becomes a significant bottleneck, causing throughput scaling to deteriorate. Furthermore, CP often necessitates a substantial number of GPUs to manage such contexts; for example, [5] utilized 256 GPUs to train a 2M context model.
>
> Therefore, OOMB methods become the preferable approach in resource-constrained settings. This includes scenarios with a limited number of GPUs or where inter-GPU communication bandwidth is restricted.
>
> [5] Chejian Xu, et al. From 128k to 4m: Efficient training of ultra-long context large language models. arXiv preprint arXiv:2504.06214, 2025.

---

> ### Author Response · Authors · 2025-11-26
>
> **[Q] The paper has limited technical novelty, as each component exists: chunk-wise training: Standard for RNNs, used by SeCO for transformers; paged KV cache: directly from vLLM; CPU offloading: common practice; Sparse attention: well-established.**
>
> [A] We sincerely thank the reviewer for their time and for raising this crucial point regarding the novelty of our technical components. We fully appreciate your observation that the individual concepts have precedents in the literature.: (i) chunk-wise processing[1], (ii) paged memory[2,3], and (iii) sparse attention[4,5].
>
> However, we respectfully submit that the novelty of OOMB lies not in the mere adoption of these components, but in their training-specific adaptation and deep synergistic integration.
>
> We would like to clarify our technical contributions and distinctiveness as follows:
>
> **1. Paged KV Cache Re-engineered for Training (Distinct from vLLM)**
>
> The reviewer correctly notes that paged memory is used in inference engines like vLLM. However, existing paged attention solutions are designed for inference and lack backpropagation support.
>
> * To enable training, we had to develop a custom Triton kernel that handles both forward and backward passes.
>
> * Unlike standard implementations, our system is designed to bypass the PyTorch autograd engine. We implemented specialized kernels where gradients are accumulated in-place using atomic operations.
>
> * We introduced a paged memory manager not just for the KV cache, but also for its gradients. This allows us to eliminate memory fragmentation when appending new key-value pairs during training, a problem inference engines do not face.
>
> **2. Synergistic Co-design**
>
> The core innovation of OOMB is the deep synergy of these techniques, which solves the memory bottleneck that no single component can address in isolation. Our system relies on 6 distinct interactions between components to achieve efficiency. These are not generic combinations but specific co-designs:
>
> * **Leveraging Locality (Chunk-wise Training $\leftrightarrow$ Sparse Attention)**: Standard sparse attention often struggles with retrieval overhead. In our design, sparse attention leverages the intra-chunk locality inherent in chunk-wise training. Since queries within a chunk exhibit similar attention patterns, we can maximize the efficiency of the retrieval process, a benefit that would be lost in standard training paradigms.
>
> * **Dynamic Memory Management (Chunk-wise Training $\leftrightarrow$ Paged KV Cache)**: Unlike inference where sequence length might be static or predictable, chunk-wise training causes the KV cache to grow dynamically. This necessitates our custom paged management system to prevent the severe memory fragmentation and expensive reallocation costs that would occur with standard contiguous memory tensors.
> * **Achieving Near Constant Memory Overhead (Chunk-wise Training $\leftrightarrow$ KV Cache Offload)**: While chunk-wise training reduces activation memory, the KV cache still grows linearly. Our system resolves this by offloading the linearly growing KV cache, which acts as the hidden state between chunks. This interaction is the key mechanism that allows OOMB to yield a near constant memory footprint throughout the training process.
>
> * **Minimizing Communication (Sparse Attention $\leftrightarrow$ KV Cache Offload)**: Standard offloading is bandwidth-intensive. However, our system utilizes sparse attention to enable the selective offloading of only essential data. By retrieving only the relevant pages needed for the current computation, we significantly reduce the communication overhead compared to retrieving the full context, making offloading viable even for massive sequences.
>
> * **Granular Transfer Efficiency (KV Cache Offload $\leftrightarrow$ Paged KV Cache)**: Finally, the efficiency of offloading relies on the memory structure. The page-level granularity of the paged KV cache enables the selective transfer (in form of contiguous memory) of specific data blocks via DMA. This makes the offloading mechanism both flexible and highly efficient, avoiding the latency spikes associated with transferring large, monolithic tensors.
>
>
> **3. Empirical Breakthrough**
>
>
> The ultimate proof of this system-level novelty is in the results.
>
>
>  1. _Efficiency_. For every additional 10K tokens, our end-to-end memory overhead increases by a mere 10MB.
>
>  2. _Capability_. This integration allows training a 4M-token context on a single H200 GPU. Achieving this with standard Context Parallelism would require large clusters.
>
>
>  We hope this clarifies that OOMB is not a simple assembly of existing parts, but a specialized training system that overcomes the specific barriers of gradient computation and memory management in ultra-long contexts. We have revised the text to explicitly highlight these distinctions based on your valuable feedback.

---

> ### Author Response · Authors · 2025-11-26
>
> [1] Wenhao Li, et al. Training Long-Context LLMs Efficiently via Chunk-wise Optimization. arXiv preprint arXiv:2505.16710, 2025.
>
> [2] Woosuk Kwon, et al. Efficient Memory Management for Large Language Model Serving with PagedAttention. arXiv preprint arXiv:2309.06180, 2023.
>
> [3] Lianmin Zheng, et al. SGLang: Efficient Execution of Structured Language Model Programs. arXiv preprint arXiv:2312.07104, 2024.
>
> [4] Jingyang Yuan, et al. Native Sparse Attention: Hardware-Aligned and Natively Trainable Sparse Attention. arXiv preprint arXiv:2502.11089, 2025.
>
> [5] Enzhe Lu, et al. MoBA: Mixture of Block Attention for Long-Context LLMs. arXiv preprint arXiv:2502.13189, 2025.

---

### Official Review · Reviewer_dZ2Y · 2025-10-21

**Soundness:** 3
**Presentation:** 4
**Contribution:** 3
**Rating:** 6
**Confidence:** 3

**Summary:**

This paper introduces OOMB, a highly memory-efficient training system designed to overcome the prohibitive GPU memory overhead associated with training Large Language Models (LLMs) on million-token contexts. The core of the system is a chunk-recurrent training framework that employs on-the-fly activation recomputation, achieving a constant ($\mathcal{O}(1)$) memory footprint for activations. To manage the remaining bottleneck of the linearly growing KV cache, OOMB integrates a suite of synergistic optimizations: a paged memory manager for both the KV cache and its gradients, asynchronous CPU offloading, and page-level sparse attention. The authors demonstrate that their system can train a 7B model with a 4-million-token context on a single H200 GPU, a significant advance in resource efficiency.

**Strengths:**

- While the core modules used in OOMB (e.g., chunking, paged memory, CPU offload, sparse attention) are existing concepts, the paper does an excellent job of integrating them into a cohesive and interdependent system.

- The paper provides a comprehensive set of experiments to validate the system's efficiency. The performance benchmarks against standard parallel training and state-of-the-art context parallelism methods are convincing. Furthermore, ablation studies on chunk size and scalability tests up to 8 million tokens effectively demonstrate the system's robustness and practicality.

- The OOMB framework is designed not only for single-GPU scenarios but also supports multi-GPU training via Tensor Parallelism. This flexibility significantly increases the practical value of the work for a wider range of researchers and institutions.

**Weaknesses:**

- Although the authors analyze the performance of sparse attention, I still have concerns about its impact on model quality. A key missing piece of analysis is an ablation study on the page size. The representative vector for each key page is calculated by averaging. It is plausible that a larger page size would average out more details, reducing the distinctiveness between different page-representative vectors and thus degrading the quality of the Top-K retrieval. This seems like a critical hyperparameter that was not explored.

- The training loss curve for the 1M context length in Figure 7 (top-left) shows some instability, especially for the larger budget configuration. The authors briefly attribute this to the learning rate and limitations in positional encoding extrapolation. This explanation is not sufficient to be fully convinced. Could this instability be an artifact of the sparse approximation itself at such an extreme scale? Further analysis is needed to clarify this point.

**Questions:**

1.  The KV cache is also a form of activation. The OOMB framework discards other intermediate activations but retains the KV cache to be passed between chunks . Is the primary reason for this special treatment to ensure the attention mechanism has access to the full history for a mathematically correct forward pass, while other activations can be recomputed locally within a chunk without affecting the model's final output?
2.  In Table 4a, the results for 16K, 64K, and 256K contexts generally show that the loss decreases as the sparse attention budget increases, approaching the performance of dense attention. However, for the 1M context, the loss is higher with larger budgets (4K+8192 and 4K+32768) than with smaller ones. This trend is counter-intuitive. Could the authors provide a potential explanation for this behavior?

---

> ### Author Response · Authors · 2025-11-18
>
> **[Q] A key missing piece of analysis is an ablation study on the page size.**
>
> [A] Thank you for this insightful question; it clearly demonstrates your expertise in sparse attention. You are correct that a large page size can cause over-smoothing, which is a common issue for page-level sparse attention methods. We selected 128 in our paper because, on the H200, smaller page sizes (or GPU threads per block) failed to fully utilize the computational resources.
>
> As you suggested, we ran an ablation study on smaller page sizes using Qwen2.5-7B (with an 8K local window and 8192 token budget). The latency results are below:
>
>
> | Page Size | Ckpt | Offload | 64K | 128K | 256K |
> | ----------- | ----| ---- | ----  | ----  | ----  |
> | 32 | w/ | w/ | 56.3 | 154.8 | 382.9 (187%) |
> | 64 | w/ | w/ | 43.9 | 105.9 | 270.2 (132%) |
> | 128 (baseline) | w/ | w/ | 41.7 | 94.0 | 204.7 (100%) |
>
> We note that when attempting to increase the page size from 128 to 256, we encountered the following error:
>
> ```
> triton,runtime.errors.PTXASError: PTXAS error: Internal Triton PTx codegen error
> ```
>
> Therefore, we were unable to include those results. Nevertheless, these findings confirm that your analysis is perfectly accurate. A smaller page size does indeed reduce system efficiency. This is likely due to sub-optimal optimizations in our custom Triton kernel, or it may be an inherent limitation common to all page-level sparse attention methods.
>
> ---
>
> ---
>
> ---
>
> **[Q] The training loss curve for the 1M context length in Figure 7 (top-left) shows some instability. Could this instability be an artifact of the sparse approximation itself at such an extreme scale? Further analysis is needed to clarify this point.**
>
> [A] Thank you again for your keen observation. We acknowledge that we are not experts in sparse attention; our implementation was a naive approach based on NSA and MoBA.
>
> We posit that the retrieval mechanism of sparse attention is indeed highly likely to cause this training instability. Because the retrieval strategy is overly simplistic, as the context length increases, a large number of false positive tokens are retrieved. This injects significant noise into the system and, consequently, contributes to the instability observed during training.
>
> Furthermore, the reasoning in our paper for the attribution in the paper was based on our empirical finding that this instability can be eliminated by reducing the learning rate and the AdamW $\\beta\_2$ parameter. (Given a batch size of 1, the default 5e-5 learning rate and 0.98 $\\beta\_2$ are arguably high, especially for full-parameter fine-tuning). However, to maintain experimental uniformity and adhere closely to default conditions, we did not adjust these parameters beyond their default values.
>
> ---
>
> ---
>
> ---
>
> **[Q] In Table 4a, the results for 16K, 64K, and 256K contexts generally show that the loss decreases as the sparse attention budget increases, approaching the performance of dense attention. However, for the 1M context, the loss is higher with larger budgets (4K+8192 and 4K+32768) than with smaller ones.**
>
> [A] We agree with you that this is indeed anomalous. When the context window exceeds 128K, we observe many counter-intuitive phenomena, from gradient estimation to the loss itself. We have faithfully reported these details in the paper, even though the underlying cause for this phenomenon remains undetermined.
>
> A potential hypothesis is that our naive sparse attention mechanism is suboptimal. As the budget increases, it may inadvertently select numerous redundant and low-relevance blocks, thereby injecting noise and diluting the effective attention.

---

> > ### Comment · Reviewer_dZ2Y · 2025-11-24
> >
> > Thank you for your detailed rebuttal. After considering the additional information you provided, I have decided to maintain the current positive score.

---

### Official Review · Reviewer_ZwBd · 2025-10-31

**Soundness:** 3
**Presentation:** 3
**Contribution:** 3
**Rating:** 6
**Confidence:** 5

**Summary:**

Training large language models (LLMs) on long contexts is constrained by exorbitant GPU memory overhead, primarily due to the linear growth of memory usage with sequence length. This paper proposes a memory-efficient training system named OOMB, which adopts a scheme combining a "block-wise cyclic training framework" with "on-the-fly activation recomputation". To efficiently manage the KV cache, OOMB incorporates a set of collaborative optimization strategies: (1) a paged memory manager for both the KV cache and its gradients to eliminate fragmentation; (2) asynchronous CPU offloading to hide data transfer latency; (3) page-level sparse attention to reduce both computational complexity and communication overhead. The synergy of these techniques yields exceptional efficiency.

**Strengths:**

1. Paged KV Cache and Gradient Management. This paper introduces a paged memory manager for the key-value (KV) cache and its gradients to eliminate exorbitant memory operation overhead, while alleviating memory fragmentation when appending new KV pairs.
2. Specialized Kernels. This paper implement custom operators that execute all operations on the KV cache and its gradients directly, making them opaque to PyTorch’s autograd system. This avoids storing the KV cache as an activation and allows for direct gradient accumulation.
2. Asynchronous KV Cache CPU Offloading. This paper designed an asynchronous mechanism to preemptively offload the KV cache and its gradients, whose memory grows with context length, to CPU memory. The resulting data transfer latency is effectively masked by computation.
3. Page-level Sparse Attention. This technique not only reduces the complexity of attention computation but also minimizes the communication overhead introduced by the offloading mechanism to the greatest extent.

**Weaknesses:**

1. The description of the workflow of the OOMB training system lacks clarity. Could pseudocode be provided for illustration, similar to FlashAttention?
2. The experiments are only conducted on small model, and validation on larger or more diverse models is anticipated.

**Questions:**

Refer to Weaknesses

---

> ### Author Response · Authors · 2025-11-18
>
> **[Q] The description of the workflow of the OOMB training system lacks clarity. Could pseudocode be provided for illustration, similar to FlashAttention?**
>
> [A] Thank you for your feedback. We agree that pseudocode significantly enhances the clarity of our method. Accordingly, we have provided pseudocode below to illustrate our approach at three levels: 1) the kernel, 2) the attention module, and 3) the training pipeline. We have strived to make it as clear and readable as possible. We are happy to clarify any further details. For a complete implementation, please also refer to the code in our anonymous repository.
>
> ---
>
> ---
>
> ---
>
> **[Q] The experiments are only conducted on small model, and validation on larger or more diverse models is anticipated.**
>
> [A] Thank you for your insightful suggestion. Following your recommendation, we have conducted additional efficiency tests on the 14B model. First, we estimated the theoretical Flops for the 7B and 14B models at different context lengths:
>
>
> |  | 64K | 128K | 256K |
> | --- | --- | --- | --- |
> | Qwen2.5-7B | 2.479 PFlops | 8.25 PFlops | 29.64 PFlops |
> | Qwen2.5-14B | 5.718 PFlops | 19.48 PFlops | 71.19 PFlops |
> | Ratio | 2.30x | 2.36x | 2.40x |
>
> Next, we tested the time and memory overhead for OOMB Dense and OOMB Sparse (12K+32768) on the 14B model:
>
> | Time (s) / Iter | Ckpt | Offload | 64K | 128K | 256K |
> | ------- | ----- | ---- | ----- | -------| ------|
> | 14B + OOMB Dense [4K] | yes | yes | 178.8 | 637.9 | 2393.4 |
> | 14B + OOMB Sparse [12K + 32768] | yes | yes | 158.3 | 438.0 | 977.1 |
>
> | Peak Memory (MB) | Ckpt | Offload | 64K | 128K | 256K |
> | ------- | ----- | -------- | --- | ----- | ---- |
> | 14B + OOMB Dense [4K]  | yes | yes | 63283 | 64372  | 66422 |
> | 14B + OOMB Sparse [12K + 32768] | yes | yes | 73792 | 73836 | 73938 |
>
> Based on these results, we can see that the 14B model exhibits the same memory scalability as the 7B model. In terms of time efficiency, for OOMB+dense attention, the 14B model's training time is approximately 2.42x that of the 7B model, which perfectly aligns with the theoretical Flops gap. For OOMB+sparse attention, the 7B and 14B models achieve 2.33x and 2.45x speedups (vs. dense), respectively. This demonstrates that the acceleration from OOMB Sparse Attention remains effective on larger-scale models.

---

> ### Author Response · Authors · 2025-11-18
> **Algorithm 1: Sparse Paged Attention Kernel**
>
> *Algorithm 1: Sparse Paged Attention Kernel*
>
> *Require:*
> * Matrices $\mathbf{Q} \in \mathbb{R}^{N_{q} \times d}$ (Query for current chunk) in HBM.
> * $PageTable$ (T): Lookup table in HBM for $K_j, V_j$ page addresses.
> * $SelectionTable$ (ST): Retrieval table in HBM, mapping $Q_i \to ST_i$ (Top-K indices).
> * $LocalIndices$: List of local K/V page indices for the current chunk, in HBM.
> * On-chip SRAM of size $M$.
>
> *Set block sizes:*
>
> 1.  $B\_r$: Row block size for Q.
>
> 2.  $B\_c$: Column block size for K/V (i.e., `page_size`).
>
> *Initialize:*
>
> 3.  $\mathbf{O} = \mathbf{0} \in \mathbb{R}^{N_{q} \times d}$, $l = (-\infty) \in \mathbb{R}^{N_{q}}$, $m = (-\infty) \in \mathbb{R}^{N_{q}}$ in HBM.
>
> 4.  Divide $\mathbf{Q}$ into $T\_r = \\lceil N\_q / B\_r \\rceil$ blocks $\\mathbf{Q}\_1, \\dots, \\mathbf{Q}\_{T\_r}$.
>
> 5. Divide $\\mathbf{O}$, $l$, $m$ into $T_r$ blocks: $(\\mathbf{O}\_1, \\dots, \\mathbf{O}\_{T\_r})$, $(l\_1, \\dots, l\_{T\_r})$, $(m\_1, \\dots, m\_{T\_r})$.
>
> *Main Loop:*
>
> 6.  *for* $1 \le i \le T_r$ *do* (Parallel over all $Q$ blocks $Q_i$)
>
>     1.  Load $\mathbf{Q}_i$ from HBM to on-chip SRAM.
>
>     2.  Initialize $O_i = \mathbf{0}$, $l_i = -\infty$, $m_i = -\infty$ on-chip.
>
>     3.  Load $ST_i$ (block $i$'s Top-K index list) from HBM.
>
>     4.  // Loop 1: Process retrieved historical K/V blocks (Top-K) //
>
>     5.  *for* $page\\_index$ *in* $ST_i$ *do*
>
>         1.  Load $\mathbf{K}_j, \mathbf{V}_j$ from HBM (using $PageTable[page\\_index]$) to on-chip SRAM.
>
>         2.  On chip, compute $\mathbf{S}_{ij} = \mathbf{Q}_i \mathbf{K}_j^T \in \mathbb{R}^{B_r \times B_c}$.
>
>         3.  On chip, compute $m\_{ij} = \\text{rowmax}(\\mathbf{S}\_{ij})$, $\\tilde{\\mathbf{P}}\_{ij} = \\exp(\\mathbf{S}\_{ij} - m\_{ij})$, $\\tilde{l}\_{ij} = \\text{rowsum}(\\tilde{\\mathbf{P}}\_{ij})$.
>
>         4.  On chip, compute $m_i^{\text{new}} = \max(m_i, m_{ij})$.
>
>         5.  On chip, compute $l_i^{\text{new}} = e^{m_i - m_i^{\text{new}}} l_i + e^{m_{ij} - m_i^{\text{new}}} \tilde{l}_{ij}$.
>
>         6.  On chip, update $\\mathbf{O}\_i \\leftarrow \\text{diag}(l\_i^{\\text{new}})^{-1} (\\text{diag}(e^{m\_i - m\_i^{\text{new}}} l\_i) \\mathbf{O}\_i + e^{m\_{ij} - m\_i^{\\text{new}}} \\tilde{\\mathbf{P}}\_{ij} \\mathbf{V}\_j)$.
>
>         7.  Update $l_i \leftarrow l_i^{\text{new}}$, $m_i \leftarrow m_i^{\text{new}}$.
>
>     6.  *end for*
>
>     7.  // Loop 2: Process local window K/V blocks (Causal) //
>
>     8.  *for* $page\\_index$ *in* $LocalIndices$ *do*
>
>         1.  Load $\mathbf{K}_j, \mathbf{V}_j$ from HBM (using $PageTable[page\\_index]$) to on-chip SRAM.
>
>         2.  Similar as loop1.
>
>     9.  *end for*
>
>     10. Write $\mathbf{O}_i$ from on-chip SRAM to HBM.
>
>     11. (Write $l_i, m_i$ to HBM for backward pass).
> 7.  *end for*
>
> 8.  *Return* $\mathbf{O}$.

---

> ### Author Response · Authors · 2025-11-18
> **Algorithm 2: Sparse Attention Forward**
>
> *Algorithm 2: Sparse Attention Forward*
>
> *Require:*
> * $\mathbf{X} \in \mathbb{R}^{N_{q} \times D}$ (Current chunk hidden states)
> * $\mathcal{C}$ (Sparse KV Cache Manager)
> * $\mathbf{W}_q, \mathbf{W}_k, \mathbf{W}_v, \mathbf{W}_o$ (Projection matrices)
> * $L_{past} \leftarrow \mathcal{C}.NumKV$ (Number of tokens currently in cache)
> * $S \in \{1, 2\}$ (Stage: 1=Inference, 2=Training)
>
> *Procedure:*
>
> 1.  $\mathcal{C}.Visit(S)$ // Mark cache access for the current layer and stage
> 2.
> 3.  // 1. Query Projection
> 4.  $\mathbf{Q} \leftarrow \mathbf{X} \mathbf{W}_q$ // Project hidden states to Query
> 5.
> 6.  // 2. Trigger Asynchronous Retrieval
> 7.  $\mathcal{C}.Select(\mathbf{Q}, S)$ // Use $\mathbf{Q}$ to find and start fetching relevant sparse K/V pages
> 8.
> 9.  // 3. K/V Projections (Concurrent with Step 2)
> 10. $\mathbf{K} \leftarrow \mathbf{X} \mathbf{W}_k$ // Project hidden states to Key
> 11. $\mathbf{V} \leftarrow \mathbf{X} \mathbf{W}_v$ // Project hidden states to Value
> 12.
> 13. // 4. Update Cache with Current Chunk
> 14. $\mathcal{C}.Update(\mathbf{K}, \mathbf{V}, S)$ // Add current chunk's K/V to the cache
> 15.
> 16. // 5. Compute Attention (Waits for Retrieval)
> 17. $\mathbf{A} \leftarrow \text{SparsePagedFlashAttn}(\mathbf{Q}, \mathbf{K}, \mathbf{V}, \mathcal{C})$ // $\mathcal{C}$ provides retrieved + current K/V
> 18. $\mathbf{O} \leftarrow \mathbf{A} \mathbf{W}_o$ // Final output projection
> 19.
> 20. *return* $\mathbf{O}$

---

> ### Author Response · Authors · 2025-11-18
> **Algorithm 3: Block-wise Sparse Training Pipeline**
>
> *Algorithm 3: Block-wise Sparse Training Pipeline*
>
> *Require:*
> * $\mathcal{M}$ (LLM)
> * $\mathbf{X}_{full}$ (Full input sequence)
> * $\mathbf{Y}_{full}$ (Full label sequence)
> * $L$ (Number of model layers)
> * $N_h$ (Number of KV heads, tensor-parallel adjusted)
> * $B$ (Batch size)
> * $B_s$ (Block size / Chunk size)
> * $P_s$ (Page size for KV Cache)
> * $P_b$ (Page budget for sparse retrieval)
>
> *Procedure:*
>
> 1.  // 1. Initialize Sparse Paged KV Cache
>
> 2.
>
> 3.  $\mathcal{C} \leftarrow \text{InitializeSparseCache}(layers=L, bsz=B, page\_size=P_s, heads=N_h, budget=P_b)$
>
> 4.
>
> 5.  // 2. Chunk Inputs and Labels
>
> 6.  $\\mathbf{X}\_{chunks} \\leftarrow \\text{Chunkize}(\\mathbf{X}\_{full}, size=B\_s)$
>
> 7.  $\\mathbf{Y}\_{chunks} \\leftarrow \\text{Chunkize}(\\mathbf{Y}\_{full}, size=B\_s)$
>
> 8.  $T \leftarrow \text{length}(\mathbf{X}_{chunks})$ // Total number of chunks
>
> 9.
>
> 10. // 3. Main Training Loop
>
> 11. $TotalLoss \leftarrow 0$
>
> 12. *for* $t \leftarrow 1$ *to* $T$ *do*
>
> 13. $\\quad$ $\\mathbf{X}_t \\leftarrow \\mathbf{X}\_{chunks}[t]$ // Get current input chunk
>
> 14. $\\quad$ $\\mathbf{Y}_t \\leftarrow \\mathbf{Y}\_{chunks}[t]$ // Get current label chunk
>
> 15.
>
> 16. $\\quad$ // Forward pass
>
> 17. $\\quad$ $\\mathbf{Logits}\_t \\leftarrow \\text{ModelForward}(\\mathcal{M}, \\mathbf{X}\_t, \\mathcal{C})$
>
> 18.
>
> 19. $\\quad$ // Loss and Backward
>
> 20. $\\quad$ $Loss\_t \\leftarrow \\text{CalculateLoss}(\\mathbf{Logits}\_t, \\mathbf{Y}\_t)$
>
> 21. $\\quad$ $\\text{ComputeGradients}(Loss\_t)$
>
> 22. $\\quad$ $TotalLoss \\leftarrow TotalLoss + Loss_t$
>
> 23.
>
> 24. $\\quad$ // Cache $\\mathcal{C}$ persists and grows; sparse retrieval is handled internally
>
> 25. *end for*
>
> 26.
>
> 27. *return* $TotalLoss$

---

### Official Review · Reviewer_TNui · 2025-11-01

**Soundness:** 3
**Presentation:** 3
**Contribution:** 3
**Rating:** 8
**Confidence:** 3

**Summary:**

This paper introduces OOMB (Out Of the Memory Barrier) to train LLM efficiently with ultra-long contexts up to 4–8 million tokens using a single GPU. For GPU memory usage, activations and KV caches scale linearly with sequence length. OOMB combines chunk-recurrent training with a series of memory and compute optimizations that make activation memory constant and KV cache growth manageable.

Chunk-Recurrent Training: it splits long sequences into smaller chunks processed serially. Activation recomputation keeps activation memory constant to sequence length. Shifts the main bottleneck from activations to the KV cache.

Paged KV cache and gradient management: it proposed a custom paged memory manager for KV caches and gradients, accumulating gradients directly on GPU memory to reduce I/O and fragmentation.

Asynchronous KV cache offloading: Transfers KV caches between GPU and CPU memory asynchronously using pinned memory and DMA

Page-level sparse attention: introduces top-K page retrieval for dense models (e.g., Qwen2.5) and native sparse attention for GPT-OSS

**Strengths:**

originality
It showcased the effectiveness of paged KV cache manager that support forward and backward passes, combined with activation recomputation and asynchronous offloading. OOMB innovates on single-GPU training feasibility for million-token contexts. It's quite different to mainstream approaches

Quality
The paper has solid benchmarking results across context lengths, chunk sizes, and sparse retrieval budgets. Comparing to FlashAttention and Ring Flash Attention, improvements in runtime and memory are consistent and reproducible. It has clear formulations for chunk-wise training, sparse attention retrieval, and ablation studies up to 8M-token contexts. It also openly discussed latency tradeoffs with dependency on CPU–GPU bandwidth

Clarity: Figures 1–3 and 6–9 effectively illustrate the system architecture and memory scaling behavior. The writing style is clear and professional, with minimal ambiguity.

Significance: The work addresses memory constraints in long-context training. It's critical for long-tail users with limited GPU resource. Achieving 4M-token context training on a single GPU is a concrete result that could democratize access to long-context modeling. The system offers a practical path forward for future foundation model training

**Weaknesses:**

Whether baselines are state-of-the-art: The authors state (Section 5.3) that “for more optimal sparse attention strategies, we refer readers to the established literature,” which underscores that the sparse attention component remains under-developed. The analysis only reports gradient approximation errors (Fig. 7) without investigating why the sparse retrieval patterns emerge


Limited discussion of evaluation results: Section 6 acknowledges that “chunk-wise serial processing introduces latency,” but the magnitude of this penalty relative to GPU compute budget or wall-clock training time is unclear.

**Questions:**

It would be great to include a breakdown analysis of time spent on recomputation vs. data transfer vs. forward/backward compute.

Clarify what “O(1)” precisely means in practice (e.g., per-chunk constant, independent of total context but dependent on hidden size).

---

> ### Author Response · Authors · 2025-11-18
>
> **[Q] The authors state (Section 5.3) that “for more optimal sparse attention strategies, we refer readers to the established literature,” which underscores that the sparse attention component remains under-developed. The analysis only reports gradient approximation errors (Fig. 7) without investigating why the sparse retrieval patterns emerge**
>
> [A] Thank you for this insightful comment. You've correctly identified that our focus was not on optimizing the sparse attention strategy itself, which is why we referred to existing literature such as [1].
>
> Regarding sparse attention, we have identified several approaches compatible with our framework [2][3][4][5]. These methods incorporate various top-k retrieval strategies and have been found to achieve performance comparable to dense attention. Importantly, these methods are ready for integration into the OOMB training framework, given that our retrieval strategy is fully implemented using PyTorch.
>
>
> [1] Yichuan Deng, et al. How Sparse Attention Approximates Exact Attention? Your Attention is Naturally $n^c$ - Sparse. arXiv preprint arXiv:2404.02690, 2024.
>
> [2] Jingyang Yuan, et al. Native Sparse Attention: Hardware-Aligned and Natively Trainable Sparse Attention. In ACL, 2025.
>
> [3] Enzhe Lu, et al. MoBA: Mixture of Block Attention for Long-Context LLMs. arXiv preprint arXiv:2502.13189, 2025.
>
> [4] Ruyi Xu, et al. XAttention: Block Sparse Attention with Anti-diagonal Scoring. In ICML, 2025.
>
> [5] Yizhao Gao, et al. SeerAttention: Learning Intrinsic Sparse Attention in Your LLMs. arXiv preprint arXiv:2410.13276, 2025.
>
> ---
>
> ---
>
> ---
>
> **[Q] It would be great to include a breakdown analysis of time spent on recomputation vs. data transfer vs. forward/backward compute.**
>
> [A] Thank you for this valuable suggestion. As you noted, a detailed breakdown of the time efficiency is crucial. We now provide the time breakdown for the extra computation and data transfer.
>
> | Additional Forward (no grad, sole source of extra Flops) | Ckpt | Offload | 64K | 128K | 256K |
> | --- | --- |  --- | --- | --- | --- |
> | Dense | w/o | w/ | 17.5% | 18.7% | 17.2% |
> | Sparse (8K + 8192) | w/o | w/ | 16% | 15.4% | 15.8% |
> | Sparse (12K + 32768) |  w/o | w/ | 15.6% | 14.1% | 14.6% |
>
> Across our experiments, the recomputation overhead from OOMB held stable at 14-18% of the total training time. This confirms that the extra overhead is comparable to that of standard gradient checkpointing, which is used in modern LLM training by default.
>
> Next, we analyzed the time overhead specifically from the asynchronous KV cache data transfer:
>
> | KV Offload (data transfer) | Ckpt | Offload | 64K | 128K | 256K |
> | --- | --- | --- | --- |  --- | --- |
> | Dense | w/o | w/ | 2.5% | 0% | 0% |
> | Sparse (8K + 8192) | w/o | w/ | 5.4% | 8.3% | 10.2% |
> | Sparse (12K + 32768) | w/o | w/ | 1.1% | 0% | 2.9% |
>
> Specifically, for dense attention and sparse attention with a big enough token budget, the overhead from KV cache offloading is largely masked by the computation, thus incurring little additional time cost.
>
> ---
>
> ---
>
> ---
>
> **[Q] Clarify what “O(1)” precisely means in practice (e.g., per-chunk constant, independent of total context but dependent on hidden size).**
>
> [A] Thank you again for your effort. To clarify, the $\\mathcal{O}(1)$ complexity refers to the memory overhead required by the PyTorch autograd system. Technically, it refers to the memory for storing the cached forward activations needed for backpropagation. We have fixed our paper with this detailed clarification.

---

### Author Response · Authors · 2025-11-29

Dear AC,

Thank you for coordinating the review process. We would like to take this opportunity to clarify the factual errors found in the review provided by Reviewer qtG7 (rating 4).

Despite our detailed clarification during the discussion phase, we have not yet received a response from this reviewer. Therefore, we sincerely request the AC to examine the following objective facts that contradict the reviewer's claims:

1. **Factual error regarding Figure 7 (top-left):** The reviewer claimed that the loss fluctuates between 1.2 and 4.7 at 250 steps. However, the data in Figure 7 clearly shows that the loss is significantly below 2.6 at this point. Furthermore, the _upper limit of the y-axis in this figure is 4.0_. It is unclear how the reviewer analyzed data values that would seemingly exceed the chart's visible range or misread the clear trend lines.

2. **Factual error regarding Table 2:** The reviewer stated that, based on Table 2, our method encounters out-of-memory errors with dense attention when the context length exceeds 256K. There is no data in Table 2 to support this claim. On the contrary, the table shows that increasing context from 128K to 256K only increases memory usage from 35GB to 37GB, which is merely ~1/4 of the H200 GPU's capacity. We have provided additional experiments in our response demonstrating that our method can scale up to 4M context length without OOM under dense attention.

Given these factual inaccuracies and the lack of further engagement from the reviewer, we earnestly hope the AC will consider these objective clarifications during the final decision-making process.

---

Best regards,

Authors

---

### Meta-Review · Area_Chair_taH9 · 2025-12-04

**Summary:**

Training large language models (LLMs) on long contexts is constrained by exorbitant GPU memory overhead, primarily due to the linear growth of memory usage with sequence length. This paper proposes a memory-efficient training system named OOMB, which adopts a scheme combining a "block-wise cyclic training framework" with "on-the-fly activation recomputation".

**Reviewer Concerns:**

Most reviewers tend to accept it. Most concerns have been well addressed by authors.

**Reviewer Scores:**

I do not think reviewers will change their scores.

---

### Decision · Program_Chairs · 2026-01-26

Accept (Poster)